# Comparative evaluation of two autotransfusion devices in a 72h survival swine model of surgically induced controlled splenic bleeding

Axelle Castelli[1], Chloé Libaud[2], Benoît Decouture[2], Marine Bruneau[2], Mallorie Depond[2], Patricia Forest-Villegas[2], Olivier Gauthier[1,3], Audrey Lafragette[1], Gwenola Touzot-Jourde [1,4,5]¤ *

1 CRIP, Center for Research and Preclinical Investigation, Oniris Nantes Atlantic College of Veterinary Medicine, Food Science and Engineering, Nantes, France, 2 i-SEP, Nantes, France, 3 Regenerative Medicine and Skeleton, INSERM, ONIRIS, Nantes University, UMR 1229- RMeS, Nantes, France, 4 Ecole Nationale Vétérinaire d'Alfort, ACAP3, Goustranville, France, 5 Ecole Nationale Vétérinaire d'Alfort, Maisons-Alfort, France

¤Current address: EnvA Campus Normand-CIRALE, Normandie Equine Vallée, Goustranville, France
* gwenola.touzot-jourde@vet-alfort.fr

## Abstract

Autotransfusion is a key strategy in hemorrhagic surgical procedures, reducing risks like disease transmission and immunosuppression due to allogenic transfusion. While conventional devices efficiently process red blood cells by centrifugation, they don't address complications requiring additional platelet transfusions. The innovative same™ device (i-SEP, France), utilizing hollow-fiber filtration, preserves both red blood cells and platelets without damaging cell integrity. This study designed as a prospective randomized controlled trial compared cell concentration and washout performances of two autotransfusion devices, a preclinical version of same™, the conventional centrifugation-based Xtra® (LivaNova, UK); and effects of retransfusion versus no transfusion in 21 Yucatan minipigs submitted to a surgically induced controlled splenic bleeding. Animals were divided into three groups (no-transfusion *control* group, *same* group and *xtra* group) and monitored postoperatively for 72 hours during which serial animal blood samples were collected for hematologic, biochemical and coagulation analyses and tests. Both autotransfusion devices showed high performances in red blood cell yields and concentrations, with a significant superiority of same™ device to preserve platelets. Animals from *same* and *xtra* groups retrieved similar rheological profiles and maintained a normal hematocrit compared to no-transfusion control animals. Coagulation profiles stayed within normal range in all groups. No adverse event on animals nor post-mortem sign of thrombosis were identified on autotransfused animals. The same™ device that can therefore be considered as an interesting alternative to conventional centrifugation-based devices. Further experiments are needed to provide evidence of platelets autotransfusion benefits in massive hemorrhagic procedures.

**Data availability statement:** All relevant data are within the article and its Supporting information files.

**Funding:** This study was funded by BPi France (French Public Bank for Investments, https://www.bpifrance.com/) through the PSPC-402492 grant (BPI grants for structuring competitive research and development). The grant was awarded to the consortium Isep/Oniris/Université de Rennes, which includes all the authors. The funder provided support in the form of salaries for authors [BD, CL, MB, PF-V] and contributed to covering the costs of animal experimentation. However, the funder had no additional role in study design, data collection and analysis, decision to publish, or preparation of the manuscript. The specific roles of these authors are detailed in the 'Author Contributions' section.

**Competing interests:** We have read the journal's policy, and the authors of this article declare the following competing interests: Chloé Libaud (CL) is employed as a research and development engineer by i-SEP (Nantes, France). Dr. Benoit Decouture (BD) was employed as a project manager by i-SEP (Nantes, France). Dr. Mallorie Depond (MD) is currently employed as a project manager by i-SEP (Nantes, France), having joined the company after the data acquisition and completion of the study. Marine Bruneau (MB) was employed as a research and development engineer by i-SEP (Nantes, France). Dr. Patricia Forest-Villegas (PFV) is currently employed as scientific director by i-SEP (Nantes, France). The authors confirm that the employment of some co-authors by i-SEP does not alter adherence to PLOS ONE policies on data and material sharing. These affiliations have not influenced the objectivity or integrity of the research presented. All data generated during the study are fully available, and all relevant data have been incorporated into the article. Additionally, the evaluated technology is patented by the manufacturer i-SEP, and the commercial version of the system has been available on the European market since July 2022.

## Introduction

Autotransfusion is an increasingly-used attractive patient blood management strategy. Red blood cell (RBC) autotransfusion allows to reduce the need for allogenic blood transfusions (donor) in numerous surgical procedures (cardiac, orthopedic, visceral, vascular, obstetric) [1–3]. Benefits of autologous transfusion lay in the absence of infectious disease transmission, transfusion reactions and immunosuppression [4–8], while not carrying allogenic transfusion limits like is donor's availability and high processing costs [9–12]. Autotransfusion is an important pillar in the patient blood management strategy and the use of an autotransfusion device is recommended by international guidelines during procedures in which blood loss is anticipated [13,14]. Conventional autotransfusion devices process the collected blood mainly by centrifugation and provide a red blood cell concentrate for patient transfusion while removing platelets and coagulation factors with the potential of leading to dilutional coagulopathy if large blood volumes are treated and reinfused to the patient [15–17]. Resulting thrombocytopenia increases the need for platelet transfusion that is associated with an increase in postoperative complications including infections and increased length of hospital stay [14,18–21]. These disadvantages point out the need for a device able to simultaneously wash and concentrate RBC and platelets while preserving cell integrity.

The Xtra® device (LivaNova, London, UK) is one of the gold-standard cell salvage systems used in perioperative conditions for autologous blood recovery [22–24]. This device collects blood, washes and concentrates mainly red blood cells (RBC) by centrifugation within the intra-operative period [25].

The new autotransfusion medical device based on a hollow-fiber filtration technology (same™ by i-SEP, Nantes, France) has been shown in an *in vitro* study to efficiently wash and concentrate RBC as well as platelets without significant impact on cell integrity and function [26,27]. Indeed, RBC and platelet functionalities were preserved while leukocytes did not show abnormal activation nor cell death. The same™ device has then been evaluated in an *in vivo* study through a swine model reproducing two clinicals conditions of surgically induced blood loss: cardiac and visceral blood loss [27]. Pigs reinfused with same™ device treated blood had minimal variations in their whole blood count, hematocrit, hemoglobin and red blood cell concentrations during the surgical procedure and the 72h postoperative follow-up time. The reinfusion did not lead to any postoperative adverse events. No hypo- nor hyper-coagulable state was evidenced and postmortem histological examination did not reveal any thrombotic lesion [27]. The study collecting *in vivo* parameters confirmed results obtained in the previous *in vitro* study by Mansour et al. [26].

The present study aimed at prospectively comparing two autotransfusion devices: the filtration-based same™ device and the centrifugation-based Xtra® device in an *in vivo* porcine model of surgically-induced abdominal controlled hemorrhage, with a targeted blood-loss reaching 15% of the animal total blood volume [28], with a 72-hour survival. The aims were to compare device performances for blood treatment and document autotransfusion effects on animals with the blood treated by

each device as well as animal recovery against a no-transfusion control group. This study also documents effects on hematology, coagulation and animal recovery including post-transfusion reactions and complications, comparatively to a no-transfusion control group.

## Materials and methods

This preclinical study was conducted in accordance with the regulations governing good laboratory practice: The Organization for Economic Co-Operation and Development Principles of Good Laboratory Practice [29] and the United States Food and Drug Administration Good Laboratory Practice for Non-Clinical Laboratory Studies (FDA-CFR 21 Part 58) [30]. The animal study protocol was run according to the European Community Guidelines for the care and use of laboratory animals (2010/63/UE) following approval by the Pays de La Loire ethical committee for animal experimentation and authorization by the French Ministry of Higher Education, Research and Innovation (Apafis number #11079-2017071317574824v1). All procedures were performed in an animal approved facility (Oniris Experimental Surgical Plateform, accreditation number C-44-271 by Loire Atlantique prefectoral order N° 2014-DDPP-132) and under guidance of the facility animal welfare body.

### Study design, autotransfusion devices and treatment group allocation

The study was designed as a prospective randomized controlled trial using a previously described animal model of visceral blood loss by controlled splenic bleeding induced surgically in minipigs [27]. Animals were randomly allocated to 3 treatment groups: 2 autotransfused groups with either device: same™ (2019 preclinical version, i-SEP, France) – named *same* group for the *in vivo* part of the study, or Xtra˚ (LivaNova, UK) – named *xtra* group for the *in vivo* part of the study, and a group without transfusion after hemorrhage – named *control* group in *in vivo* study, with inclusion of 7 animals per group. The number of animals required was determined by a power calculation detailed in the supporting information, S1 File. The shed blood of control animals was randomly allocated to be treated either by the preclinical same™ device or the commercialized Xtra˚ device in order to increase the total number of treated bloods per device from 7 to 10 for performance evaluation. The final product obtained from control animal blood was not reinfused and therefore discarded.

### Blood collection and autotransfusion treatment programs evaluation

The cell salvage treatment was performed in accordance with the manufacturer's instructions for use and the recommended settings for processing and washing. Using the Xtra˚ device, the blood was processed in the kit X/125 mL size bowl using the device's built-in Intraoperative Standard Program (Pstd protocol) in Automatic Mode. Using the same™ device, the blood was processed in ST0300FR set with Standard mode. Detailed process of same™ device is described in previous published *in vitro* and *in vivo* studies [26,27]. Each autotransfusion devices were used to collect about 850-1000mL of diluted shed blood by animal from the surgical field, allowing to proceed to at least two consecutive treatment cycles, named cycle 1 and cycle 2. Blood samples were collected for analysis before and after each treatment cycle, in the collecting reservoir (TR) and in the transfusion bag (TB) respectively.

Qualitative and quantitative parameters were used to evaluate and compare the performances of the autotransfusion systems in washing and concentrating collected blood (RBC numeration and yield, hematocrit, hemoglobin concentration, heparin, free hemoglobin and other plasma components washout) following AABB recommendations and conventional practices [26,27], along with complementary parameters as platelet concentration and yield. Performances were calculated by formulas detailed in the supporting information, S1 File.

### Animals and surgical model of abdominal controlled blood loss

Twenty-one 14- to 22-month-old female, weighing 41.6 to 60.5 kg Yucatan minipigs were included in this study. Each animal was individually identified by an ear tag (since a few days after birth) to guarantee traceability. Detailed animal care,

**Table 1. Expected performances for criteria.**

| Parameters | Criteria | Ref. |
|---|---|---|
| *Quality of the blood to be reinfused – unwanted substances clearance* | | |
| Anticoagulant (heparin) concentration (IU/mL) | ≤0.5 | a |
| Anticoagulant clearance (%) | >90% | a |
| Hemolysis rate (%) | ≤0.8% | b |
| Free hemoglobin washout (%) | 90% | a |
| Other plasma components washout (%) | | b |
| *Quantification of blood elements to transfuse and device performances* | | |
| Red blood cells yield (%) | > 80% | b |
| Final Hematocrit (%) | 45% < Ht > 65% | a |
| Platelet yield (%) | NA | NA |
| Platelets Concentration ($10^6$/μL) | NA | NA |

a. Based on AABB recommendations [31].

b. Based on literature and conventional practices [32–34].

NA, not applicable.

anesthesia and analgesia protocol as well as surgical model of abdominal controlled blood loss have been described in [27] and in the supporting information, S1 File. Briefly, after a week of acclimation in an enriched environment, animals were immobilized by an intramuscular injection. During anesthesia maintained with a balanced technique under controlled ventilation, arterial and central venous accesses were gained before surgically-induced controlled bleeding and auto-transfusion. To counteract known porcine hypercoagulability [35–37] and to facilitate blood loss and collection, a low dose of unfractionated heparin (Heparin sodium 5 000 IU/mL, 25 IU/kg IV) was administered just before initiating the splenic injury. This dosage has been used in previous studies of porcine intra-abdominal hemorrhage and has shown to bring the porcine coagulation profile within values found in humans [27,38,39]. ACT measurements (Activated Clotting Time, via Medtronic ACT II Coagulation Timer) were done before heparin administration, repeated during the blood collection time (target ACT 90–130 seconds) and just before transfusion to ensure a return to baseline ACT value. Following a midline laparotomy, multiple lesions were created into the splenic parenchyma using digitoclastic technique and an additional lesion was done by severing the gastroepiploic vein. Spontaneous bleeding was let to drip into the abdominal cavity and then aspirated through the suction line (depression kept under 200 mbar to minimize hemolysis) [39] into the autotrans-fusion device blood collection reservoir (BCR). The target volume for blood loss was set to 15% of the estimated animal blood volume (61–68 ml/kg, calculus method in S1 File) [40] in order to reach the first stage of hemorrhagic shock induc-ing substantial hemodynamic compromises [28,41]. Taking into consideration that the spleen is responsible for platelet clearance from blood circulation and that maintaining splenic function postoperatively is necessary to assess physiological response to autotransfusion [42], surgically-induced splenic bleeding was controlled by performing a hemi-splenectomy using an endoscopic surgical stapler with 60 mm-long cartridges (Echelon Flex™ Endopath®, Ethicon Endosurgery, John-son and Johnson). Left gastroepiploic vessels were then coagulated using a tissue fusion device (Atlas Ligasure™, Covid-ien, Medtronic). After the end of surgery and autotransfusion, the animals were allowed to recover and were euthanized at the end of a 72h follow-up period after transfusion completion. A timeline synopsis synthetizes the schedule of the operat-ing theater and the blood samples management (Fig 1). Anesthesia depth and blood pressure support (intravenous fluid replacement and catecholamines) were adjusted to the animal response to acute blood loss to maintain a mean blood pressure above 60 mmHg while waiting for autotransfusion and if retransfusion did not stabilize sufficiently hemodynamic parameters. Postoperative care comprising analgesia and supportive therapy was adapted to each individual recovery during a 72-hour follow-up period post-hemorrhage until euthanasia. Animal welfare as well as anesthesia procedure,

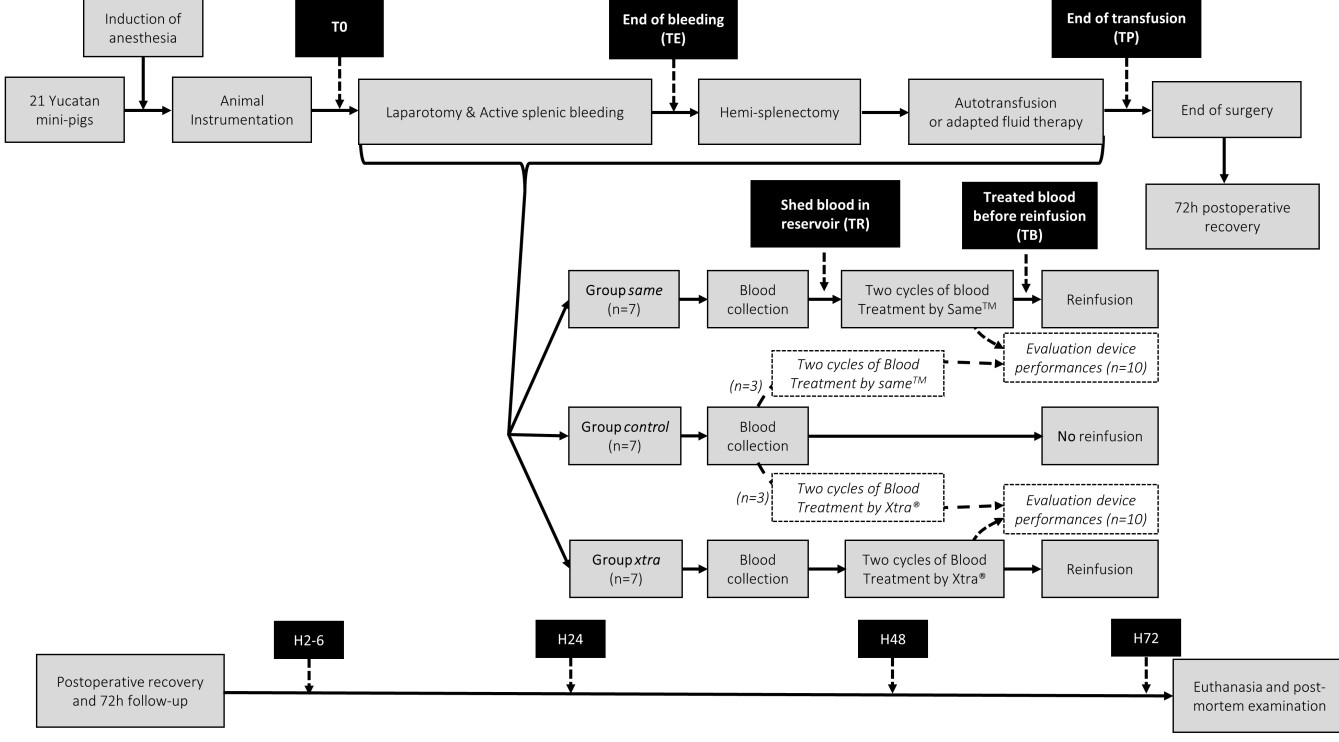

**Fig 1. Timeline synopsis of surgery and blood samples management.** Times of sampling were before surgery once the central venous catheter was in place (T0), at the end of bleeding (TE), at the end of the transfusion (TP), then postoperatively between 2 and 6 hours (H2), at 24 (H24), 48 (H48) and 72 hours post-transfusion (H72). Additional blood samples were obtained from the BCR (TR) and from the transfusion bag (TB) before the reinfusion started.

postoperative care and postoperative follow-up are detailed in S1 File. Euthanasia and post-mortem examination were geared to assess thrombogenic risk as described in the supporting information, S1 File.

## Laboratory testing

Blood samples were collected from before surgery to the end of the 72-hour follow up to monitor possible variations in hematologic and coagulation parameters. Times of sampling are described in Fig 1. Total blood volume collected was refined by establishing the minimal volume needed for laboratory testing ahead of the experiment. All blood samplings were performed by gentle aspiration through the central venous catheter which allowed stress-free repetitive procedures on the minipigs postoperatively. Additional blood samples were obtained from the BCR (TR) and from the transfusion bag (TB) before the reinfusion started. Blood analyses consisted of complete blood count (CBC) including RBC, WBC and platelet count, hematocrit and total hemoglobin (Procyte Dx Hematology Analyzer, IDEXX, Hoofddorp, The Netherlands), plasma free hemoglobin concentration measurement for the calculation of the hemolysis washout (Plasma/Low Hb pho-tometer, HemoCue AB, Sweden), biochemical assays: sodium, chloride, potassium (VetLyte, IDEXX, The Netherlands); calcium, phosphate, glucose, lactate dehydrogenase, total protein, albumin, triglycerides, lactate (RX Daytona VetTest, IDEXX, The Netherlands); non-esterified fatty acids (NEFA FS, Diasys), TNF-alpha (Porcine TNF-alpha ELISA kit, R&D Systems), pig-map (Pig MAP (major acute phase protein) ELISA kit, FineTest), haptoglobin (haptoglobin assay, Tridelta PHASE), c-reactive protein (porcine CRP assay kit, Tridelta Phase), D-Dimers (D-Dimer ELISA kit, Technozym), plasma heparin concentration measurement through anti-Xa activity assay (Heparin standard-HNF, HemosIL Liquid anti-Xa,

ACLTOP, IL, Werfen, France). Coagulation status was assessed by thromboelastometry with ROTEM non heparin sensitive EXTEM test (ROTEM delta, Werfen, France). Analyzed parameters were clotting time (CT; s), maximum clotting firmness (MCF; mm), AUC (a.u.), α angle (°) and clot lysis at 30 minutes (Ly30; %). Classical coagulation tests (fibrinogen, prothrombin time and activated partial thromboplastin time) were also performed (Start 4°, STAGO, France).

## Statistical analysis

Statistical analyses of continuous variables were performed with Multiple Mann-Whitney test and demographic data was compared between groups with a Kruskal-Wallis for non-parametric data. All the data are presented as median (min-max). All statistical tests were two-tailed, a ROUT (Prism adapted Dixon's Q) test was performed with 1% FDR to exclude outlier values. A p-value of less than 0.05 was considered statistically significant and represented by stars in tables and figures where * is ≤0.05, ** is ≤0.01, *** ≤0.005, **** ≤0.001 and ns for not significative. The statistical analysis was conducted using GraphPad Prism software 10.2.2 (GraphPad Software, Boston, Massachusetts USA).

## Results

### Medical devices treatment performances

Collected blood allowed for at least two consecutive treatment cycles for each animal and each device. Cycle 1 and 2 were independently analyzed. Due to larger collected volume, a third cycle was performed on three occasions with each device (3 out of 10 for Xtra° device and 3 out of 10 for same™ device). However, blood volumes were incomplete and non-representative, so data of third cycles are not shown.

A median diluted blood volume of 1090mL (882–1342) per animal was treated with the Xtra° device, allowing to retrieve a median final concentrated blood volume of 245mL (238–432). A median diluted blood volume of 824mL (631–1148) per animal was treated by same™, allowing to obtain a median final concentrated blood volume of 205mL (110–417). Diluted blood volume was slightly higher for Xtra° compared to the same™ (* p = 0.0108) but the concentrated blood volume was not significantly different despite a tendency (ns p = 0.06). Diluted blood from the blood collection reservoir (BCR) before treatment (TR) and concentrated blood in the transfusion bag after treatment (TB) were compared for multiple cell yields and plasma components washouts. All data are summarized in Tables 2 and 3.

### RBC concentrations and yield

Initial hematocrits and RBC concentrations of diluted shed blood were not different between groups in TR for cycle 1, and the treatment process achieved a concentrated blood in cycle 1 with similar hematocrits and RBC concentration for Xtra° and same™. For cycle 2, initial hematocrit and RBC concentration in TR were lower for Xtra° than same™. Cycle 2 concentrated blood in TB showed a lower hematocrit and RBC concentration for Xtra° than same™. However, median hematocrits in TB were within the recommended 45–65% range [32] for both devices. Detailed results are presented in the Fig 2 and Table 2. RBC recovery performance evaluated by RBC yield were similar between devices at each cycle as presented in Fig 2.

### Platelets concentrations and yield

Initial platelet concentration in TR was not different between devices for cycle 1. Filtration-based same™ device allowed to concentrate platelets to achieve a high platelet concentration in TB, resulting in a significantly six times higher platelet concentration and nine times higher yield for same™ device than Xtra° device in TB. At the cycle 2, despite a significantly lower initial concentration of platelets for same™ device compared to Xtra°, treatment cycle resulted in a significantly four times higher concentration and seven times higher yield of platelets by same™ device than Xtra° device in TB. Detailed results are presented in the Fig 2 and Table 2.

**Table 2. Autotransfusion devices performances in cell concentrations for the two successive treatment cycles.**

| Devices | | Cycle 1 | | Difference between devices | Cycle 2 | | Difference between devices |
|---|---|---|---|---|---|---|---|
| | | Xtra® (n = 10) | same™ (n = 10) | p-value | Xtra® (n = 10) | same™ (n = 10) | p-value |
| RBC concentration (10⁶/μL) | TR | 2.4 | 2.2 | *ns* | 2.6 | **3.1** | ** |
| | | (1.8–4.2) | (1.6–3.2) | | (2.0–4.1) | (2.0–4.0) | |
| | TB | 9.0 | 7.6 | *ns* | 8.6 | **9.3** | * |
| | | (8.2–9.6) | (4.7–9.1) | | (6.5–9.7) | (6.1–10.1) | |
| Hematocrit (%) | TR | 15 | 16 | *ns* | 14 | **18** | ** |
| | | (11–28) | (13–26) | | (11–19) | (13–25) | |
| | TB | 58 | 54 | *ns* | 49 | **57** | * |
| | | (55–61) | (42–61) | | (31–58) | (46–65) | |
| RBC yield (%) | | 82.4 | 76.7 | *ns* | 86.1 | 95.5 | *ns* |
| | | (44.8–83.9) | (58.4–89.4) | | (78.9–106.6) | (56.4–113.2) | |
| Platelet Concentration (10⁶/μL) | TR | 58 | 43 | *ns* | **56** | 40 | ** |
| | | (34–156) | (26–71) | | (38–84) | (33–71) | |
| | TB | 14 | **85** | **** | 18 | **78** | *** |
| | | (8–41) | (50–113) | | (12–36) | (60–139) | |
| Platelet Yield (%) | | 5.1 | **47.7** | **** | 7.9 | **59** | *** |
| | | (3.2–12.4) | (32.7–132.6) | | (5.1–13.4) | (48.4–80.0) | |

The two devices were compared for cycle 1 and 2. Results are expressed as median (min-max). TR: in reservoir, before treatment cycle. TB: in transfusion bag, after treatment cycle. p-value significance is represented by stars where * is ≤0.05, ** is ≤0.01, *** ≤0.005, **** ≤0.001 and ns for not significative. Best results are highlighted in bold.

## Washout performances

**Heparin concentration and washout.** Results are detailed in Fig 3 and Table 3. For cycle 1, heparin concentration was similar between same™ and Xtra® devices in TR, decreased in TB after treatment and was significantly lower for same™ than Xtra®, meaning a significantly higher washout by same™ device than Xtra® device. For cycle 2, heparin concentration in TR was significantly higher for Xtra® than same™, decreased in TB and was not significantly different between devices, meaning a washout not significantly different between devices.

**Free hemoglobin concentration and washout.** Plasmatic free hemoglobin concentration was significantly lower in TR for same™ device than Xtra® device at cycle 1 and cycle 2. In TB, free hemoglobin concentration at cycle 1 was not different between groups but was significantly lower at cycle 2 for Xtra® device than same™ with 0.97 g/dL (0.65–1.16) vs 1.31 g/dL (1.1–87) respectively, *** p = 0.0001. The calculated washout of free hemoglobin was higher with Xtra® device than same™ device for cycle 1 and cycle 2 (Fig 3 and Table 3).

**Plasmatic components washouts.** Washouts of other plasmatic components were calculated independently for cycle 1 and cycle 2 for both devices and almost all parameters were similarly washed with no statistical difference (detailed in Table 3). Briefly creatinine, haptoglobin and lactate washouts reached at least an 80% rate for both devices with no significant differences. Potassium, calcium, phosphate, albumin, total protein, glucose, NEFA, LDH, Pig-MAP and CRP washouts reached at least an 90% rate for both devices with no significant differences. Only two exceptions of significant differences were noted: potassium and triglycerides. Potassium washouts were above 90% for both devices but Xtra® device showed a better performance than same™ device at cycle 1 (median Potassium washout of 94.7% vs 91.5% respectively, p < 0.0001) and at cycle 2 (median Potassium washout of 93% vs 91.7% respectively, p = 0.0141). Triglycerides washouts were above 80% for both devices but Xtra® device showed a better performance than same™

**Table 3. Autotransfusion devices performances in cell concentrations for the two successive treatment cycles.**

| Devices | | Cycle 1 | | Difference between devices | Cycle 2 | | Difference between devices |
|---|---|---|---|---|---|---|---|
| | | Xtra® (n = 10) | same™ (n = 10) | p-value | Xtra® (n = 10) | same™ (n = 10) | p-value |
| Heparin concentration (IU/mL) | TR | 8.6 | 7.8 | ns | **8.4** | 6.6 | ** |
| | | (3.5–9.5) | (6.1–9.7) | | (7.6–9.3) | (4.9–8.1) | |
| | TB | 0.94 | **0.31** | *** | 0.52 | 0.11 | ns |
| | | (0.28–1.44) | (0–0.68) | | (0–1.12) | (0–3.02) | |
| Heparin washout (%) | | 97.8 | **99.4** | ** | 98.5 | 99.6 | ns |
| | | (96.8–99.2) | (96.4–100) | | (97.3–100) | (85.8–100) | |
| Free Hb concentration (g/dL) | TR | 0.65 | **0.36** | *** | 0.7 | **0.51** | ** |
| | | (0.37–1.81) | (0.18–0.63) | | (0.37–1.03) | (0.3–0.76) | |
| | TB | 0.65 | 1.30 | ns | **0.97** | 1.31 | *** |
| | | (0.52–0.95) | (0.38–2.01) | | (0.65–1.16) | (1.1–87) | |
| Free Hb washout (%) | | **92.4** | 61.6 | **** | **84** | 61.6 | *** |
| | | (84.6–94.8) | (−18.1–87.5) | | (51.4–91.2) | (−14.1–73.9) | |
| Potassium washout (%) | | **94.7** | 91.5 | **** | **93** | 91.7 | * |
| | | (92.7–96.6) | (72.9–93.6) | | (90.7–94.9) | (70–93.8) | |
| Calcium washout (%) | | 96.8 | 97.4 | ns | 96.8 | 97.3 | ns |
| | | (95.1–97.6) | (86.7–98.5) | | (94.3–97.3) | (72.2–98.1) | |
| Phosphate washout (%) | | 95.8 | 96.2 | ns | 96.4 | 96.9 | ns |
| | | (46.7–98.5) | (83.8–98.4) | | (59.1–98.3) | (85.3–98.6) | |
| Creatinine washout (%) | | 85.4 | 84.1 | ns | 83.4 | 86.3 | ns |
| | | (79.7–92.9) | (69.5–89.7) | | (77.4–86.4) | (69.2–91.7) | |
| Albumin washout (%) | | 94.1 | 92.7 | ns | 92.6 | 91.5 | ns |
| | | (81.3–96.8) | (69.5–94.7) | | (80.3–94.6) | (71.4–93.8) | |
| Protein washout (%) | | 94.6 | 94 | ns | 93.5 | 90.4 | ns |
| | | (63.3–97.6) | (47–96.1) | | (63.6–96.1) | (63.1–96.4) | |
| Glucose washout (%) | | 97.6 | 96.2 | ns | 96.6 | 96.5 | ns |
| | | (76.1–98.4) | (69.5–98.6) | | (75.3–98.1) | (72.2–98.6) | |
| Triglycerides washout (%) | | **91.5** | 85.6 | ** | **90.3** | 81.3 | * |
| | | (67.4–93.5) | (38.4–91.7) | | (62–91.9) | (58.2–90.1) | |
| NEFA washout (%) | | 94.6 | 95.6 | ns | 90.8 | 92.8 | ns |
| | | (81.3–97.4) | (92.4–97.6) | | (79.9–96) | (65.3–97.5) | |
| Haptoglobin washout (%) | | 96.6 | 94.5 | ns | 96.35 | 88.4 | ns |
| | | (−481.1–99.8) | (−1545–99.7) | | (−338.7–99.9) | (−5735–99.2) | |
| LDH washout (%) | | 99.7 | 99.5 | ns | 99.6 | 99.1 | ns |
| | | (64.1–100) | (69.5–100) | | (42.2–100) | (−1841–100) | |
| Lactate washout (%) | | 88.9 | 87.8 | ns | 88.9 | 89.9 | ns |
| | | (86.8–93.2) | (77.2–92.4) | | (85.1–94.2) | (77.8–92.6) | |
| TNFalpha washout (%) | | 97.8 | 100 | ns | 79.85 | 87.3 | ns |
| | | (14.7–100) | (−11.2–100) | | (−127.6–100) | (10.9–100) | |
| PigMAP washout (%) | | 93.7 | 96.1 | ns | 93.5 | 92.7 | ns |
| | | (89.9–98.3) | (91.3–99.1) | | (90.2–100) | (72.5–98.9) | |
| CRP washout (%) | | 95.3 | 93.5 | ns | 89.6 | 94.9 | ns |
| | | (75.2–100) | (−1.7–99.4) | | (76.9–99.7) | (90.4–99.9) | |

The two devices are compared for cycle 1 and 2. Results are expressed as median (min-max). TR: in reservoir, before treatment cycle. TB: in transfusion bag, after treatment cycle. p-value significance is represented by stars where * is ≤0.05, ** is ≤ 0.01, *** ≤0.005, **** ≤0.001 and ns for not significative. Best results are highlighted in bold.

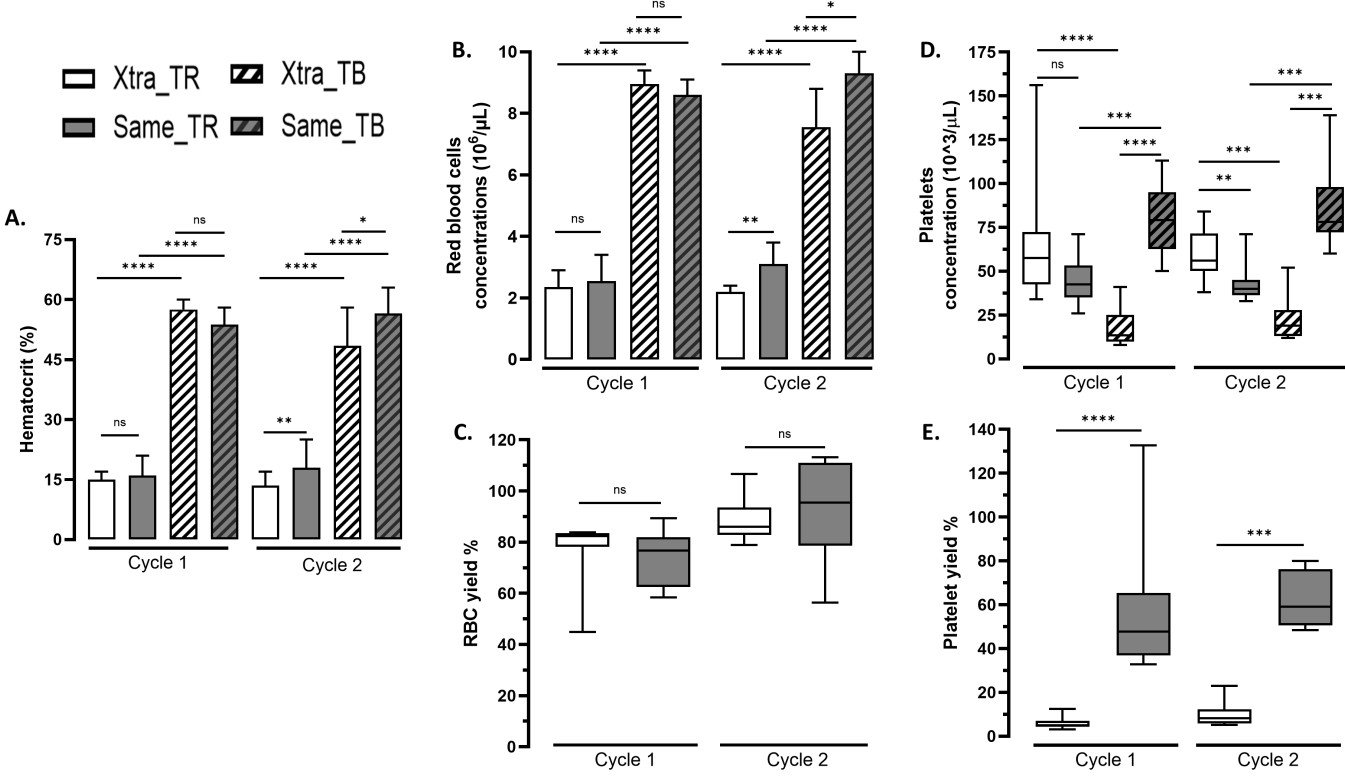

**Fig 2. Comparative performances of the two autotransfusion devices.** In white Xtra® device, in gray same™ device, with stripes in transfusion bag after blood treatment, data represent median and error bars for min-max for each cycle. **A.** Hematocrit (%) in input (reservoir) and output (transfusion bag). **B.** RBC concentrations ($10^6$/µL) in input (reservoir) and output (transfusion bag). **C.** Calculated RBC yield (%). **D.** Platelet concentrations ($10^3$/µL) in input (reservoir) and output (transfusion bag). **E.** Calculated platelet yield (%).

device at cycle 1 (median triglycerides washout of 91.5% vs 85.6% respectively, p = 0.0064) and at cycle 2 (median triglycerides washout of 90.3% vs 81.3% respectively, p = 0.032).

## Data on animals

After device performance evaluations on processed blood, the *in vivo* part of the study compared pigs with autotransfusion, *xtra* and *same* groups, or no-transfusion *control* group during a 72-h follow-up (n = 7 for each animal group). There was no significant difference between the three groups for anesthesia duration, bleeding speed, transfusion time delay from the end of bleeding to the beginning of transfusion and total ringer lactate volume administered during anesthesia (Table 4).

In the no-transfusion *control* group, 57% of the animals (4/7) required advanced intravenous (IV) fluid management during anesthesia with a bolus of colloid solution and 29% (2/7) received an additional bolus of hypertonic saline with the colloids to maintain hemodynamic stability. One animal in the *same* group (14%) received a bolus of hypertonic saline. During immediate recovery, a total of 8 minipigs had slow recovery and required IV fluid administration, representing 43% in the *control* group (during 4h for 2 animals and 8h for 2 animals), 29% in *xtra* group for 4 to 5 hours and 43% in *same* group for 3 hours. Two animals, 1 *control* and 1 *xtra*, necessitated administration of nasal oxygen after extubation for a few hours. All had resumed normal behavior and appetite the following morning except one *xtra* animal that had decreased appetite, increased sleeping time and required additional fluid and energy support during the day.

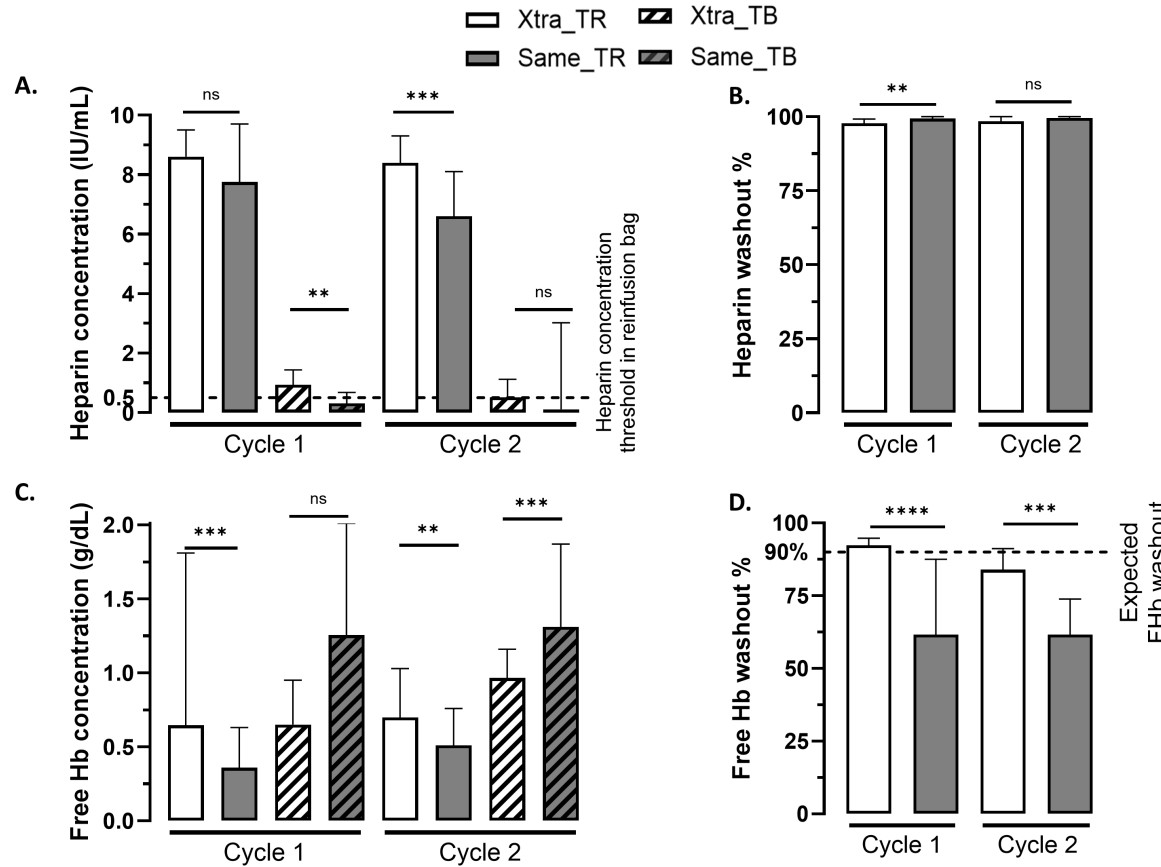

**Fig 3. Comparative washout of the two autotransfusion devices.** In white Xtra® device, in gray same™ device, with stripes for transfusion bag after blood treatment, data represent median and error bar for max for each cycle. **A.** Heparin concentration (IU/mL) in input (blood collection reservoir, no stripes) and output (transfusion bag, with stripes). Threshold represented at 0.5IU/mL is corresponding to AABB requirement of maximal remaining heparin concentration in reinfusion bag. **B.** Calculated heparin washout (%). **C.** Free hemoglobin concentration (g/dL) in input (reservoir, no stripes) and output (transfusion bag, with stripes). **D.** Calculated free hemoglobin washout (%). Above a 90%-threshold of FHb washout, AABB considers a good performance.

**Hematologic follow-up.** Complete hematological follow-up was performed with complete blood cell counts (CBC) during anesthesia before the surgery (T0), at the end of bleeding (TE), at the end of the transfusion (TP), then postoperatively at the four time points of the follow-up (H2, H24, H48 and H72). Table 5 presents selected parameters and detailed CBC is displayed in the supporting information, S1 Table.

Right after retransfusion for *xtra* and *same* groups (TP), RBC count, hematocrit and hemoglobin were significantly higher than the no-transfusion *control* group (blood sample taken at the end of anesthesia). Parameters stayed statistically higher in the reinfused groups at all time of follow-up for *xtra* group (H2, H24, H48 and H72) and at H24 for *same* group compared to the *control* group. No difference in the WBC counts were identified between groups. All animals developed postoperatively a neutrophilic leukocytosis with a peak at H24 that started resolving between H48 and H72. No other significant difference between groups at matching times was found in the remaining CBC parameters (reticulocytes, platelets or among other white blood cells).

**Biochemical analysis.** Analyses on animal blood samples included concentration measurement of sodium, potassium, chlore, creatinine, albumin, total proteins, glucose, triglycerides, calcium, phosphate, and NEFA. No

**Table 4. Anesthetic management synthesis.**

| ANIMAL GROUPS | *control* group (n = 7) | *xtra* group (n = 7) | *same* group (n = 7) |
|---|---|---|---|
| **Anesthesia duration (min)** | 225 (155–300) | 195 (170–260) | 210 (160–230) |
| **Bleeding speed (mL/kg/h)** | 11.7 (5.5–32.4) | 9.7 (6.7–16.7) | 7.7 (4.8–19.5) |
| **Blood loss volume (%)** | 13.4% (8.7–26.4) | 15.2% (11.6–17.4) | 12.6% (9.2–25.2) |
| **Delay before reinfusion (min)** | No transfusion | 21 (19–26) | 23 (18–48) |
| **Intravenous crystalloid volume during anesthesia (mL/kg)** | 33 (25–36) | 29 (22–43) | 27 (19–32) |
| **Number of animals with advanced IV fluid administration post-bleeding** | 4 animals<br>Colloid solution (Voluven 6%) 2.5–4 mL/kg<br><br>2 animals of the 4 Hypertonic saline 10% 1–1.5 mL/kg | 0 animal | 1 animal<br>Hypertonic saline 10% 1 mL/kg |
| **Intravenous crystalloid volume during recovery (for 4 to 8 hrs)** | 3 animals 13–43 mL/kg | 2 animals 22–35 mL/kg | 3 animals 10–18 mL/kg |

Results are expressed as median (min-max).

significant difference between control and both groups of transfused animals were identified at the successive time points (supporting information, S1 Table). Follow-up of plasmatic concentrations of haptoglobin, LDH, lactate, used to assess hemolysis along with TNF-alpha, PigMAP and CRP, selected to assess inflammation, were also performed. A higher lactate concentration at the end of anesthesia was found for the *control* group (4.33 (3.43–4.81) mmol/L) compared to the end of transfusion (TP) for *same* group (2.95 (1.02–3.5) mmol/L, p = 0.0047) but only a tendency with the *xtra* group (2.8 (1.85–4.98) mmol/L, ns p = 0.0513). TNF-alpha concentration at H48 was significantly lower for *same* group and *xtra* group compared to *control* group (respectively 21.7 (15.8–47.8), p=0.0076 and 24.1 (8.8–103.3), p=0.0285 versus 46.6 (24.8–148.5) ng/L). TNF-alpha concentration was also significantly lower at H72 for *same* group compared to *control* group at H72 (respectively 19.9 (0–30.3) and 35.1 (17.3–135.7) ng/L, p = 0.0221). There was no statistically significant difference between *the 3* groups for the other parameters (haptoglobin, LDH, PigMAP and CRP).

**Coagulation parameters.** To specifically document the effects of platelet reinfusion provided by the same™ device, coagulation marker measurements and tests were performed: D-Dimer and fibrinogen concentrations; prothrombin time (PT), activated partial thromboplastin time (APTT), ROTEM thromboelastrometry (clotting time (CT), maximal clot firmness (MCF), α-angle, clot lysis at 30 minutes (Ly30), area under the curve (AUC)). Results are presented in the Table 6. Results did not show any abnormal response and any significant differences between groups over time.

**Necropsy observations.** Macroscopic observations during thorough necropsy examination did not show any difference between the three groups regarding undesirable thrombotic events. More details of euthanasia and examination are provided in the supporting information, S1 File.

## Discussion

The aim of this study was to compare the new filtration-based same™ device and the conventional centrifugation-based Xtra® device in a swine model of surgically induced controlled splenic bleeding. Pigs, and especially minipigs, have become a major translational research model over the last three decades and have replaced the dog as the general large animal surgical model in many areas of the world, with great acceptance by regulatory authorities [43,44]. A splenic

**Table 5. Selected hematological parameters in animals during the 72 h follow-up.**

| Parameters | Groups | T0 | TE | TP | H2 | H24 | H48 | H72 |
|---|---|---|---|---|---|---|---|---|
| **RBC count (10⁶/µL)** | control | 5.5 | 5.7 | 4.8 | 5.3 | 4.4 | 4.5 | 4.4 |
| | | (4.6–6) | (4.4–5.9) | (4–5.4) | (4.7–6) | (3.5–5.1) | (3.8–6.4) | (3.7–5) |
| | xtra | 5.1 | 5.6 | **5.8\*\*** | **6.2\*** | **5.7\*\*** | **5.6\*** | **5.0\*\*** |
| | | (4.9–6.5) | (5.2–6.1) | (5–6.3) | (5.4–6.9) | (5.2–6.1) | (4.8–5.9) | (4.9–6.1) |
| | same | 4.8 | 5.5 | **5.5\*** | 5.8 | **5.0\*** | 5.1 | 5.3 |
| | | (4.4–5.6) | (4.5–6.1) | (4.2–6.3) | (5.1–6.7) | (4.3–5.8) | (4.2–5.4) | (4.4–5.9) |
| **Hemoglobin (g/L)** | control | 111 | 113 | 95 | 105 | 85 | 93 | 91 |
| | | (90–121) | (90–120) | (82–109) | (97–124) | (73–101) | (75–126) | (76–100) |
| | xtra | 104 | 115 | **123\*\*** | **133\*** | **115\*\*** | **110\*** | **106\*** |
| | | (103–134) | (105–124) | (102–131) | (110–147) | (105–125) | (103–122) | (97–127) |
| | same | 102 | 115 | **117\*** | **121\*** | **107\*** | 110 | 112 |
| | | (92–116) | (94–125) | (88–130) | (107–139) | (90–121) | (87–114) | (95–124) |
| **Hematocrit (%)** | control | 35 | 36 | 29 | 33 | 26 | 29 | 29 |
| | | (27–39) | (28–38) | (26–34) | (30–40) | (22–31) | (23–42) | (23–32) |
| | xtra | 33 | 37 | **38\*\*** | **42\*** | **36\*\*\*** | **35\*** | **34\*** |
| | | (32–44) | (34–40) | (32–41) | (35–49) | (33–39) | (32–39) | (30–40) |
| | same | 33 | 37 | **37\*** | 39 | **32\*\*** | 34 | 34 |
| | | (28–37) | (29–40) | (27–42) | (33–45) | (28–38) | (28–37) | (30–40) |
| **Platelets (10³/µL)** | control | 313 | 269 | 250 | 275 | 240 | 261 | 315 |
| | | (141–434) | (217–379) | (189–326) | (207–409) | (140–385) | (151–380) | (225–446) |
| | xtra | 337 | 289 | 254 | 295 | 309 | 304 | 332 |
| | | (104–447) | (232–362) | (182–345) | (225–417) | (256–401) | (224–428) | (139–465) |
| | same | 422 | 338 | 333 | 336 | 319 | 311 | 344 |
| | | (239–468) | (207–390) | (218–365) | (201–388) | (217–385) | (219–361) | (265–419) |
| **WBC (10³/µL)** | control | 9.6 | 8.1 | 9.3 | 13.9 | 19.4 | 15.8 | 15.4 |
| | | (7.7–11.3) | (8.1–11.8) | (7.5–14) | (9.4–16.5) | (17.1–22.2) | (9.3–19.2) | (9.5–18.5) |
| | xtra | 9.6 | 10.6 | 10.7 | 14.7 | 18.7 | 12.9 | 12.6 |
| | | (6.7–19.2) | (7.8–13) | (7.9–16.2) | (9.8–15.6) | (14.1–24.7) | (11.3–17.8) | (9.8–16.3) |
| | same | 9.7 | 8.3 | 9.2 | 16.3 | 18.6 | 15.4 | 13.9 |
| | | (8.5–12.5) | (7.3–11.3) | (6.5–12.5) | (9.6–19.8) | (13.4–24.8) | (11.1–20) | (10–19.1) |
| **Neutrophilic granulocytes (10³/µL)** | control | 4 | 5.3 | 5.9 | 11.4 | 15.4 | 9.6 | 8.2 |
| | | (3–5.3) | (4.2–6.9) | (2.2–9.1) | (5.2–13.5) | (12.5–17.5) | (5.6–14) | (5.2–12.1) |
| | xtra | 3.3 | 5.8 | 6.4 | 10.3 | 12.9 | 7 | 7.3 |
| | | (2.8–16.9) | (3.6–9.3) | (4.6–10.2) | (6.4–12.6) | (10.7–18.7) | (6–12.3) | (5–11.3) |
| | same | 4.5 | 4.9 | 6 | 12.4 | 13.9 | 9.2 | 7.8 |
| | | (2.9–5.9) | (2.8–7) | (2.8–8.9) | (6.6–15.8) | (9.1–18.3) | (5.8–13.2) | (5.2–11.6) |

Complete blood count available in the supporting information, S1 File.

xtra (n = 7) and same (n = 7) groups were independently compared to control group (n = 7) at the corresponding time-point. Results are expressed as median (min-max). p-value significance is represented by stars where * is ≤0.05, ** is ≤0.01, *** ≤0.005, **** ≤0.001. Significant differences from control group are highlighted in bold.

T0: baseline under anesthesia once central venous catheter in place and before the surgery start, TE: end of bleeding, TP: end of transfusion, H2: between 2 and 6 hours postoperatively, H24-H48-H72: 24-, 48- and 72-hours post-transfusion.

**Table 6. Follow-up of coagulations markers (D-Dimer and fibrinogen concentrations, PT and APTT, ROTEM EXTEM).**

| Parameters | Groups | T0 | TE | TP | H2 | H24 | H48 | H72 |
|---|---|---|---|---|---|---|---|---|
| **D-Dimer concentration (µg/L)** | control | 0.0 | 22.7 | 12.3 | 0.0 | 0.0 | 0.0 | 0.0 |
| | | (0–146.5) | (0–48.2) | (0–111) | (0–73.9) | (0–104.8) | (0–10.9) | (0–36.1) |
| | xtra | 0.0 | 0.0 | 7.8 | 17.7 | 17.7 | 0.0 | 7.3 |
| | | (0–39.4) | (0–326.9) | (0–336.1) | (0–356.2) | (0–331) | (0–327.9) | (0–374.3) |
| | same | 0.0 | 10.4 | 14.8 | 0.0 | 0.0 | 4.9 | 18.0 |
| | | (0–85.7) | (0–199.6) | (0–114) | (0–149.6) | (0–57) | (0–200.8) | (0–169.9) |
| **Fibrinogen concentration (g/L)** | control | 2.04 | 1.95 | 1.29 | 1.91 | 4.30 | 5.23 | 4.95 |
| | | (1.63–2.6) | (1.72–2.04) | (1.05–1.56) | (1.34–2) | (1.72–5.46) | (4.13–6.9) | (4.19–6) |
| | xtra | 2.09 | 1.72 | 1.37 | 2.15 | 4.86 | 5.46 | 4.64 |
| | | (1.13–2.79) | (1.16–2.09) | (1–2) | (1.52–2.44) | (3.99–5.75) | (4.99–5.52) | (4.42–4.84) |
| | same | 2.28 | 1.69 | 1.94 | 1.99 | 5.11 | 5.19 | 4.67 |
| | | (1.4–3.69) | (1.42–3.02) | (1.33–2.98) | (1.45–3.55) | (4.43–6) | (4.3–5.95) | (3.56–5.88) |
| **Prothrombin time (s)** | control | 12 | 13 | 14 | 13 | 15 | 15 | 15 |
| | | (11–13) | (13–13) | (13–15) | (13–14) | (14–16) | (13–15) | (13–16) |
| | xtra | 13 | 14 | 14 | 14 | 15 | 15 | 15 |
| | | (12–15) | (12–15) | (13–18) | (12–16) | (14–16) | (15–17) | (14–18) |
| | same | 14 | 13 | 13 | 14 | 15 | 14 | 14 |
| | | (13–14) | (13–15) | (13–14) | (13–15) | (14–17) | (14–16) | (14–15) |
| **APTT (s)** | control | 10.0 | 16.0 | 19.0 | 14.0 | 17.0 | 16.0 | 14.5 |
| | | (8–15) | (14–19) | (15–22) | (13–14) | (16–18) | (14.2–18) | (10–18) |
| | xtra | 12 | 17 | 17 | 15 | 17 | 16 | 16 |
| | | (8–14) | (14–22) | (14–22) | (14–18) | (16–20) | (15–18) | (14–17) |
| | same | 14 | 16 | 15 | 13 | 17 | 18 | 15 |
| | | (13–15) | (15–18) | (14–21) | (10–16) | (14–18) | (16–21) | (14–16) |
| **CT (s)** | control | 49 | 55 | 58 | 59 | 69 | 64 | 69 |
| | | (43–72) | (51–59) | (54–78) | (51–64) | (49–82) | (61–120) | (39–80) |
| | xtra | 48 | 61 | 60 | 61 | 66 | 67 | 72 |
| | | (36–82) | (48–65) | (49–78) | (50–73) | (53–76) | (61–77) | (62–79) |
| | same | 52 | 58 | 52 | 71 | 65 | 70 | 64 |
| | | (43–71) | (50–64) | (49–58) | (58–78) | (61–74) | (65–75) | (53–74) |
| **MCF (mm)** | control | 71 | 67 | 66 | 68 | 77 | 78 | 78 |
| | | (70–75) | (62–70) | (64–70) | (67–72) | (72–78) | (71–80) | (76–79) |
| | xtra | 72 | 68 | 68 | 70 | 75 | 76 | 78 |
| | | (58–76) | (62–72) | (64–70) | (61–74) | (75–79) | (74–78) | (76–80) |
| | same | 73 | 71 | 71 | 72 | 78 | 76 | 77 |
| | | (67–80) | (62–79) | (64–76) | (63–76) | (73–79) | (74–79) | (75–82) |
| **AUC (arbitrary unit)** | control | 7049 | 6710 | 6568 | 6799 | 7661 | 7746 | 7787 |
| | | (6960–7375) | (6156–6980) | (6337–6932) | (6670–7133) | (7155–7781) | (7097–8011) | (7590–7875) |
| | xtra | 7165 | 6754 | 6721 | 6952 | 7485 | 7616 | 7742 |
| | | (5823–7585) | (6194–7183) | (6360–6951) | (6098–7342) | (7465–7854) | (7446–7723) | (7539–7955) |
| | same | 7265 | 7068 | 7028 | 7210 | 7743 | 7578 | 7677 |
| | | (6700–7891) | (6138–7869) | (6354–7531) | (6299–7597) | (7239–7850) | (7383–7925) | (7470–8134) |
| **α angle (°)** | control | 77 | 74 | 75 | 76 | 76 | 77 | 78 |
| | | (74–78) | (72–77) | (71–77) | (74–77) | (73–77) | (65–78) | (74–79) |
| | xtra | 77 | 75 | 75 | 76 | 75 | 76 | 77 |
| | | (68–80) | (72–77) | (73–76) | (65–77) | (72–78) | (71–78) | (74–80) |

*(Continued)*

**Table 6.** (Continued)

| | | | | | | | |
|---|---|---|---|---|---|---|---|
| | same | 75 | 77 | 77 | 75 | 76 | 76 | 77 |
| | | (74–80) | (70–78) | (73–78) | (70–77) | (69–78) | (73–77) | (75–78) |
| **Ly30 (%)** | control | 97 | 97 | 98 | 98 | 98 | 98 | 97 |
| | | (95–98) | (96–99) | (97–100) | (97–99) | (97–100) | (97–100) | (96–98) |
| | xtra | 98 | 98 | 98 | 99 | 98 | 97 | 97 |
| | | (95–99) | (95–99) | (96–100) | (97–100) | (96–99) | (96–99) | (94–99) |
| | same | 97 | 97 | 98 | 97 | 98 | 97 | 97 |
| | | (94–100) | (95–99) | (95–100) | (95–99) | (97–99) | (96–99) | (94–99) |

Xtra (n = 7) and same (n = 7) groups were independently compared to control group (n = 7) at the corresponding time-point. Results are expressed as median (min-max). Any significant differences were found between groups.

T0: baseline under anesthesia once central venous catheter in place and before the surgery start, TE: end of bleeding, TP: end of transfusion, H2: between 2 and 6 hours postoperatively, H24-H48-H72: 24-, 48- and 72-hours post-transfusion.

bleeding model followed by an hemisplenectomy on minipigs was used to assess device performances regarding cell concentrations and undesirable component washout and animal recovery during a 72h post-operative follow-up.

Concerning autotransfusion device performance comparison, RBC concentration was equivalent between the devices during the first cycle of treatment, but RBC yield was significantly higher for same™ device compared to Xtra® device during the second cycle. Regarding platelet concentration, same™ device performances were also superior to Xtra® device for the final platelet numeration and yield. Indeed, platelet concentration decreased in Xtra® treated blood while it significantly increased with same™. Heparin concentration in diluted blood was coherent with conventional practices and close from human heparinization in cardiac surgery [45] and heparin washout was very efficient for both devices. The blood treatment by same™ device fully respected the AABB threshold concentration of residual heparin (< 0.5 IU/mL [32]) but blood treatment by Xtra® device retrieved a slightly higher median of residual heparin concentration at cycle 1, above AABB recommendation. The filtration-based autotransfusion device demonstrates cell-salvage efficacy comparable to the gold-standard centrifugation-based device, in strong agreement with AABB guidelines [31,46].

Free hemoglobin concentrations and washouts constitute the main difference between the two devices. On one hand, the initial free hemoglobin concentration in the collecting reservoir (TR), was significantly lower for *same* group suggesting less RBC lysis induced during aspiration or in the reservoir of same™ device. On the other hand, the final free hemoglobin concentration in the transfusion bag (TB), is significantly lower for *xtra* group leading to a possible explanation that Xtra® treatment induces less RBC lysis or is able to better wash free hemoglobin. As all the other components were highly washed with same™ device during treatment cycles, it is advanced that the use of the hollow fiber filtration in conjunction with the known RBC fragility in swine species [40,42] may have resulted in more hemolysis. The study was undertaken with a preclinical version of the same™ device that has been since improved as demonstrated by a clinical study where people undergoing an on-pump cardiac surgery received an autotransfusion with the updated version of same™ showed a free hemoglobin concentration after treatment cycles within the AABB recommendations [31,47].

Multiple analyses were performed on animal blood samples during surgery and postoperative follow-up with only a few parameters showing statistically significant differences between *control* group and retransfused groups. One expected main difference was a higher RBC count, hemoglobin concentration and hematocrit in retransfused groups compared to *control* group right at the end of transfusion (corresponding to the end of anesthesia for control) and during the 72-h postoperative follow-up. Reinfusions with Xtra® and same™ devices treated blood succeeded in compensating RBC loss during bleeding. The transient neutrophilic leukocytosis observed for the three groups, is consistent with splenic surgery [48,49], and cannot be attributed to autotransfusion, as the *control* group had the same profile. As far as inflammation

and hemolysis markers are concerned, TNF-α exhibited a significantly lower concentration for the *same* group compared to *control* group. This might indicate a possible protective effect of retransfused platelets after same™ blood treatment against inflammation [50,51]. All other tested parameters were not different between conditions.

No adverse event was encountered following autotransfusion, particularly with same™ device comforting findings of the first human study [27]. As the measurement of the clotting parameters by rotational thromboelastometry did not show any abnormal response or difference between groups, it can be considered that the reinfusion of an autotransfusion blood product containing platelets did not induce any pro-coagulant effect to the animals. No *in vivo* nor post-mortem deleterious effect of retransfusion were identified with both devices and particularly considering the presence of platelets with the same™ treated blood.

The hemorrhagic shock induced in this study was not massive. Standardization of the hemorrhagic shock is difficult to obtain. The fixed volume hemorrhage model combined with a soft tissue injury [52] was chosen to comply with the variable animal body weight and the needed minimal volume to collect in the device blood collection reservoir to be able to run two treatment cycles. However, it has been shown to be more reliable than the fixed pressure model [52]. Indeed, it resulted in individual response to blood loss by variable degree and duration of hypotension, needs for fluid resuscitation and catecholamines. Lactate concentration during bleeding and transfusion showed great individual variations from being in the upper part of the normal range to highly increased. Some animals were able to maintain lactate concentration below 2 mmol/L during anesthesia and transfusion when others had increased concentration above 7 mmol/L but concentration mostly normalized in the post-operative follow-up as soon as 2 hours postoperatively confirming the mild degree of shock [52]. Adequate evaluation of autotransfusion potential side effects may be hindered in a model of severe hemorrhagic shock due to altered organic functions, decreased survival and cofounding factors brought by advanced resuscitation treatment [52]. The blood loss was significant enough to warrant, for the *control* group that was not transfused, the need for an adapted IV fluid therapy to hasten recovery and insure survival at 72h in comparison with autotransfused groups. The limited need of resuscitation did not induce any dilutional coagulopathy that would have negatively influenced the thrombogenic risk assessment of autotransfusion [52].

The controlled blood loss volume is one of the limitations of this study that may masks the beneficial effect of the platelet autotransfusion. An animal model of massive hemorrhagic shock with intense bleeding, able to induce a dilutional coagulopathy (blood loss >40–60% blood volume [52]), may highlight the autotransfused platelet benefits. Such a model imposes poor and short-term survival unless resuscitative strategies including administration of various blood products are implemented [52]. Those strategies would represent confounding factors with the benefit assessment of transfused RBC and platelets after same™ device treatment.

## Conclusions

This *in vivo* animal comparative study of controlled splenic bleeding and reinfusion allowed to describe the new autotransfusion same™ device performances in comparison with the conventional commercialized Xtra˚ device. The same™ device showed comparable performances in terms of washout and also better performances regarding cell concentrations (RBC and platelets) without any undesirable effect during the 72-h hour post-transfusion follow-up period in pigs. Animal hematological and biochemical variables, parameters assessing inflammation or coagulation states were studied and did not show major differences from the no-transfusion *control* group and both retransfused groups after Xtra˚ or same™ device blood treatment, except the benefit of RBC transfusion. Other studies should be designed to evaluate the transfusion clinical benefits and more specifically the transfusion of functional platelets during surgery especially in a massive hemorrhagic model.

## Supporting information

**S1 File. Materials and methods – additional details.**
(DOCX)

**S1 Table. Hematology and biochemistry on serial animal blood samples during the intervention and the 72h follow-up period.**
(DOCX)

## Acknowledgments

Patrice Roy for animal facilities.

## Author contributions

**Conceptualization:** Benoit Decouture, Patricia Forest-Villegas, Olivier Gauthier, Gwenola Touzot-Jourde.

**Data curation:** Axelle Castelli, Chloé Libaud, Benoit Decouture, Marine Bruneau, Patricia Forest-Villegas, Audrey Lafragette, Gwenola Touzot-Jourde.

**Formal analysis:** Axelle Castelli, Chloé Libaud, Benoit Decouture, Marine Bruneau, Mallorie Depond, Audrey Lafragette, Gwenola Touzot-Jourde.

**Funding acquisition:** Patricia Forest-Villegas.

**Investigation:** Axelle Castelli, Chloé Libaud, Benoit Decouture, Marine Bruneau, Olivier Gauthier, Audrey Lafragette, Gwenola Touzot-Jourde.

**Methodology:** Chloé Libaud, Benoit Decouture, Marine Bruneau, Mallorie Depond, Patricia Forest-Villegas, Olivier Gauthier, Audrey Lafragette, Gwenola Touzot-Jourde.

**Project administration:** Benoit Decouture, Mallorie Depond, Patricia Forest-Villegas, Olivier Gauthier.

**Resources:** Patricia Forest-Villegas.

**Software:** Chloé Libaud.

**Supervision:** Mallorie Depond, Patricia Forest-Villegas, Olivier Gauthier.

**Validation:** Patricia Forest-Villegas.

**Writing – original draft:** Axelle Castelli, Benoit Decouture, Mallorie Depond.

**Writing – review & editing:** Chloé Libaud, Benoit Decouture, Marine Bruneau, Mallorie Depond, Patricia Forest-Villegas, Olivier Gauthier, Audrey Lafragette, Gwenola Touzot-Jourde.

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
