## [Decision Letter · Decision Letter 0]

29 Jan 2025

PONE-D-24-33101Comparative evaluation of two autotransfusion devices in a 72h survival swine model of surgically induced controlled visceral blood lossPLOS ONE

Dear Dr. Touzot-Jourde,

Thank you for submitting your manuscript to PLOS ONE. After careful consideration, we feel that it has merit but does not fully meet PLOS ONE’s publication criteria as it currently stands. Therefore, we invite you to submit a revised version of the manuscript that addresses the points raised during the review process.

We look forward to receiving your revised manuscript.

Kind regards,

Stephen Emilio Njolomole, MB,BS ,MPH

Guest Editor

PLOS ONE

Journal Requirements:

2. Thank you for stating the following in the Competing Interests/Financial Disclosure* (delete as necessary) section:

“Chloé Libaud (CL) is employed as a research and development engineer by i-SEP (Nantes, France).

Dr. Benoit Decouture (BD) is currently employed as project manager by i-SEP (Nantes, France).

Dr. Mallorie Depond (MD) was employed as project manager by i-SEP (Nantes, France).

Marine Bruneau (MB) was employed as a research and development engineer by i-SEP (Nantes, France).

Dr. Patricia Forest-Villegas (PVF) is currently employed as scientific director by i-SEP (Nantes, France).

The authors confirm the fact that some of the co-authors are employed by i-SEP does not alter their adherence to PLOS ONE policies on sharing data and materials. All data generated during the study are fully available and all relevant data have been incorporated into the manuscript.”

We note that one or more of the authors are employed by a commercial company: i-SEP (Nantes, France)

3. We note that your Data Availability Statement is currently as follows: “All relevant data are within the manuscript and in Supporting Information files.”

Additional Editor Comments:

**
*Compliment*
**

The paper “Comparative evaluation of two autotransfusion devices in a 72h survival swine model of surgically induced controlled visceral blood loss“ represents an original work in its content. The paper is easy to read, with clearly defined objectives, detailed methodology. The results are clearly presented. I enjoyed reading the paper.

Good luck with the publication

**
*Compliment and Questions*
**

Well written, a model with hypovolemic shock might have shown the benefit of the better platelet recovery.

If the clinical study has already been published (this was mentioned under Discussion, about free Hb concentration), why is the preclinical study findings being published late? Was an updated version of the machine tested in preclinical model?

This study designed as a randomized controlled trial compared cell concentration and washout performances of two autotransfusion devices, a preclinical version of same™, the conventional centrifugation-based Xtra® (LivaNova, UK), with the conclusion that the same™ device showed comparable performances in terms of washout and also better performances regarding cell concentrations (RBC and platelets) without any undesirable effects.

Below are minor comments or questions:

In the abstract you wrote : Yucatan minipigs submitted to a surgically induced controlled splenic bleeding.

In the introduction: in vivo porcine model of surgically-induced abdominal controlled hemorrhage

In the methods: using a previously described animal model of controlled visceral hemorrhage by surgically induced splenic lesions in minipigs [27]. Are the published results in reference 27 the same as in this manuscript?

Did you perform the trials parallel? I ask, because if not, there would have to be a control for every run. Or the experiments are "mixed" in time. I suspect that the latter has happened.

Can you mention the type of the used heparin please?

**
*Issues to do/ to consider*
**

226 for Xtra® device and 3 out of 1O ( please insert zero instead of Capital letter O)

250 same™. Cycle 2 concentrated blood in TB showed an inferior hematocrit and RBC

251 concentration for Xtra® than same™. However, median hematocrits in TB were within

252 the recommended 45 to 65% range [30] for both devices  ( We suggest the use the word “lower” than inferior)

Please replace in the header … visceral blood loss by controlled splenic bleeding

and use in the text always controlled splenic bleeding

Please share AABB guideline referred to in this document on recommended parameters of Auto-transfused blood as pdf as we are not able to download it online due to financial restrictions.

Reviewers' comments:

Reviewer's Responses to Questions

**Comments to the Author**

1. Is the manuscript technically sound, and do the data support the conclusions?

Reviewer #1: Yes

Reviewer #2: Yes

Reviewer #3: Yes

2. Has the statistical analysis been performed appropriately and rigorously? 

Reviewer #1: Yes

Reviewer #2: Yes

Reviewer #3: Yes

3. Have the authors made all data underlying the findings in their manuscript fully available?

Reviewer #1: Yes

Reviewer #2: Yes

Reviewer #3: Yes

4. Is the manuscript presented in an intelligible fashion and written in standard English?

Reviewer #1: Yes

Reviewer #2: Yes

Reviewer #3: Yes

5. Review Comments to the Author

Reviewer #1: Well written, a model with hypovolemic shock might have shown the benefit of the better platelet recovery.

If the clinical study has already been published (this was mentioned under Discussion, about free Hb concentration), why is the preclinical study findings being published late? Was an updated version of the machine tested in preclinical model?

Reviewer #2: The paper “Comparative evaluation of two autotransfusion devices in a 72h survival swine model of surgically induced controlled visceral blood loss“ represents an original work in its content. The paper is easy to read, with clearly defined objectives, detailed methodology. The results are clearly presented. I enjoyed reading the paper.

Good luck with the publication!

Reviewer #3: This study designed as a randomized controlled trial compared cell concentration and washout performances of two autotransfusion devices, a preclinical version of same™, the conventional centrifugation-based Xtra® (LivaNova, UK), with the conclusion that the same™ device showed comparable performances in terms of washout and also better performances regarding cell concentrations (RBC and platelets) without any undesirable effects.

I have only minor comments or questions:

In the abstract you wrote : Yucatan minipigs submitted to a surgically induced controlled splenic bleeding.

In the introduction: in vivo porcine model of surgically-induced abdominal controlled hemorrhage

In the methods: using a previously described animal model of controlled visceral hemorrhage by surgically induced splenic lesions in minipigs [27]. Are the published results in reference 27 the same as in this manuscript?

Please replace in the header … visceral blood loss by controlled splenic bleeding

and use in the text always controlled splenic bleeding

Did you perform the trials parallel? I ask, because if not, there would have to be a control for every run. Or the experiments are "mixed" in time. I suspect that the latter has happened.

Can you mention the type of the used heparin please?

By the way, good work :)

6. PLOS authors have the option to publish the peer review history of their article (what does this mean? ). If published, this will include your full peer review and any attached files.

**Do you want your identity to be public for this peer review?** For information about this choice, including consent withdrawal, please see our Privacy Policy .

Reviewer #1: No

Reviewer #2: No

Reviewer #3: No

---

## [Author Response · Author response to Decision Letter 0]

4 Mar 2025

Revision Letter

PONE-D-24-33101

Comparative evaluation of two autotransfusion devices in a 72h survival swine model of surgically induced controlled splenic bleeding

PLOS ONE

Please find below our response to te academic editor and reviewers point by point

Thank you for your comments and suggestions

Response to the academic editor

Journal Requirements:

1.Please ensure that your manuscript meets PLOS ONE's style requirements, including those for file naming.

We have reviewed the style requirements and have implemented required changes.

2. Thank you for stating the following in the Competing Interests/Financial Disclosure* (delete as necessary) section:

Please find below the updated and completed funding statement

The study was funded by BPi France (French Public Bank for investments, https://www.bpifrance.com/) by the grant PSPC-402492 (BPI grants for structuring competitive research and development). The grant was awarded to the consortium Isep/Oniris/Université de Rennes (all the authors).

The funders had no role in study design, data collection and analysis, decision to

publish, or preparation of the manuscript. The funder provided support in the form of salaries for authors [BD, CL, MB, MD, PF-V] and participated in covering the cost of animal experimentations, but did not have any additional role in the study design, data collection and analysis, decision to publish, or preparation of the manuscript. The specific roles of these authors are articulated in the ‘author contributions’ section

Updated competing interests statement

Chloé Libaud (CL) is employed as a research and development engineer by i-SEP (Nantes, France).

Dr. Benoit Decouture (BD) was employed as project manager by i-SEP (Nantes, France).

Dr. Mallorie Depond (MD) is currently employed as project manager by i-SEP (Nantes, France), she joined the company after the data acquisition and completion of the study.

Marine Bruneau (MB) was employed as a research and development engineer by i-SEP (Nantes, France).

Dr. Patricia Forest-Villegas (PVF) is currently employed as scientific director by i-SEP (Nantes, France).

The authors confirm the fact that some of the co-authors are employed by i-SEP does

not alter their adherence to PLOS ONE policies on sharing data and materials. These affiliations do not influence the objectivity or integrity of the research presented in this manuscript All data generated during the study are fully available and all relevant data have been incorporated into the manuscript.

The evaluated technology is patented by the manufacturer i-SEP. The commercial version of the system evaluated in this study has been available on the European market since July 2022.

The statements have been incorporated in our cover letter.

3. We note that your Data Availability Statement is currently as follows: “All relevant data are within the manuscript and in Supporting Information files.”

Raw data files have been added to the submission of the revised manuscript

Full ethics statement has been moved from the supplemental file to the beginning of the Methods section.

The list of supplemental information files and names has been added at the end of the manuscript just before the references

The reference list has been checked.

Additional Editor Comments:

Compliment

The paper „Comparative evaluation of two autotransfusion devices in a 72h survival swine model of surgically induced controlled visceral blood loss“ represents an original work in its content. The paper is easy to read, with clearly defined objectives, detailed methodology. The results are clearly presented. I enjoyed reading the paper.

Good luck with the publication

Thank you very much for your comments

Compliment and Questions

Reviewer #1: Well written, a model with hypovolemic shock might have shown the benefit of the better platelet recovery.

If the clinical study has already been published (this was mentioned under Discussion, about free Hb concentration), why is the preclinical study findings being published late? Was an updated version of the machine tested in preclinical model?

Response to Reviewer #1:

The first clinical study, that was a non comparative study, was published after the first non-comparative preclinical study on swine (reference 27), using 2 models of blood loss (splenic bleeding and cardiac surgery with bypass). A preclinical version of the device (not completely assembled) was used for the first swine study and an updated version (finished designed used to obtain EU marketing authorization) for the first study in humans and the present study, although some software changes have been implemented (20w for the human study and 20y for the last animal study).

Reviewer #2: The paper “Comparative evaluation of two autotransfusion devices in a 72h survival swine model of surgically induced controlled visceral blood loss” represents an original work in its content. The paper is easy to read, with clearly defined objectives, detailed methodology. The results are clearly presented. I enjoyed reading the paper.

Good luck with the publication!

Response to Reviewer #2: Thank you for your encouragements

Reviewer #3: This study designed as a randomized controlled trial compared cell concentration and washout performances of two autotransfusion devices, a preclinical version of same™, the conventional centrifugation-based Xtra® (LivaNova, UK), with the conclusion that the same™ device showed comparable performances in terms of washout and also better performances regarding cell concentrations (RBC and platelets) without any undesirable effects.

Below are minor comments or questions:

Response to Reviewer #3:

In the abstract you wrote : Yucatan minipigs submitted to a surgically induced controlled splenic bleeding.

In the introduction: in vivo porcine model of surgically-induced abdominal controlled hemorrhage

Thank you for the remark. We have uniformized the wording using “surgically induced controlled splenic bleeding”.

In the methods: using a previously described animal model of controlled visceral hemorrhage by surgically induced splenic lesions in minipigs [27]. Are the published results in reference 27 the same as in this manuscript?

The study described in the reference 27 is a separate study from the present report. There is no overlap of animal use nor results between both studies.

Did you perform the trials parallel? I ask, because if not, there would have to be a control for every run. Or the experiments are "mixed" in time. I suspect that the latter has happened.

The trial described in reference 27 was run prior the comparative study presented here and focused on evaluating the device performances on two model of blood loss (splenic bleeding and cardiac surgery with bypass). The experiment on the three groups in the present study were indeed mixed in time as the treatment allocation was randomized.

Can you mention the type of the used heparin please?

Unfractionated heparin (heparin sodium 5 000 IU/mL) was used.

Issues to do/ to consider

226 for Xtra® device and 3 out of 1O ( please insert zero instead of Capital letter O)

Modification done and highlighted

250 same™. Cycle 2 concentrated blood in TB showed an inferior hematocrit and RBC

251 concentration for Xtra® than same™. However, median hematocrits in TB were within

252 the recommended 45 to 65% range [30] for both devices (We suggest the use the word “lower” than inferior)

Thank you for the suggestion

Please replace in the header … visceral blood loss by controlled splenic bleeding

and use in the text always controlled splenic bleeding

Thanks you for the remark, standardization of the wording using as suggested “ controlled splenic bleeding” has been implemented

Please share AABB guideline referred to in this document on recommended parameters of Auto-transfused blood as pdf as we are not able to download it online due to financial restrictions.

We are sorry that we are not able to provide the 2002 AABB guidelines as the paper book was lost during isep headquarter relocation 5 years ago. We have not been able to find another copy and the subsequent updated guidelines do not repeat the initial recommendations on the quality of the blood products. However we are providing with our submission two references that have used the same guidelines.

Vieira SD, da Cunha Vieira Perini F, de Sousa LCB, Buffolo E, Chaccur P, Arrais M, Jatene FB. Autologous blood salvage in cardiac surgery: clinical evaluation, efficacy and levels of residual heparin. Hematol Transfus Cell Ther. 2021 Jan-Mar;43(1):1-8. doi: 10.1016/j.htct.2019.08.005.

Buys WF, Buys M, Levin AI. Reinfusate Heparin Concentrations Produced by Two Autotransfusion Systems. J Cardiothorac Vasc Anesth. 2017 Feb;31(1):90-98. doi: 10.1053/j.jvca.2016.06.014.

Thanks again for the constructive feedback and best regards

---

## [Decision Letter · Decision Letter 1]

24 Mar 2025

Comparative evaluation of two autotransfusion devices in a 72h survival swine model of surgically induced controlled splenic bleeding

PONE-D-24-33101R1

Dear Dr. Touzot-Jourde,

We’re pleased to inform you that your manuscript has been judged scientifically suitable for publication and will be formally accepted for publication once it meets all outstanding technical requirements.

Kind regards,

Stephen Emilio Njolomole, MB,BS, MPH

Guest Editor

PLOS ONE

Additional Editor Comments (optional):

Reviewers' comments:

Reviewer's Responses to Questions

**Comments to the Author**

1. If the authors have adequately addressed your comments raised in a previous round of review and you feel that this manuscript is now acceptable for publication, you may indicate that here to bypass the “Comments to the Author” section, enter your conflict of interest statement in the “Confidential to Editor” section, and submit your "Accept" recommendation.

Reviewer #1: All comments have been addressed

Reviewer #2: (No Response)

Reviewer #3: (No Response)

2. Is the manuscript technically sound, and do the data support the conclusions?

Reviewer #1: Yes

Reviewer #2: Yes

Reviewer #3: Yes

3. Has the statistical analysis been performed appropriately and rigorously? 

Reviewer #1: Yes

Reviewer #2: N/A

Reviewer #3: Yes

4. Have the authors made all data underlying the findings in their manuscript fully available?

Reviewer #1: Yes

Reviewer #2: Yes

Reviewer #3: Yes

5. Is the manuscript presented in an intelligible fashion and written in standard English?

Reviewer #1: Yes

Reviewer #2: Yes

Reviewer #3: Yes

6. Review Comments to the Author

Reviewer #1: All comments have been addressed satisfactorily. This is a well written manuscript. Wish you Good luck!

Reviewer #2: The author answered the reviewers questions and corrected the manuscript in accordance with their suggestions.

I recommend the publication of this manuscript.

Reviewer #3: Thank you for answering my questions and considering my comments. The manuscript is well prepared and easy to read.

7. PLOS authors have the option to publish the peer review history of their article (what does this mean? ). If published, this will include your full peer review and any attached files.

**Do you want your identity to be public for this peer review?** For information about this choice, including consent withdrawal, please see our Privacy Policy .

Reviewer #1: **Yes: ** Aby Abraham

Reviewer #2: No

Reviewer #3: No

---

## [Editor Report · Acceptance letter]

PONE-D-24-33101R1

PLOS ONE

Dear Dr. Touzot-Jourde,

I'm pleased to inform you that your manuscript has been deemed suitable for publication in PLOS ONE. Congratulations! Your manuscript is now being handed over to our production team.

Kind regards,

on behalf of

Dr. Stephen Emilio Njolomole

Guest Editor

PLOS ONE